# A prospective observational study of community-acquired bacterial bloodstream infections in Metro Manila, the Philippines

**Nobuo Saito**[1,2,3], **Rontgene M. Solante**[4], **Ferdinand D. Guzman**[4], **Elizabeth O. Telan**[4], **Dorcas V. Umipig**[4], **Joy P. Calayo**[4], **Carina H. Frayco**[4], **Jezreel C. Lazaro**[4], **Maricel R. Ribo**[4], **Alexis Q. Dimapilis**[4], **Virginia O. Dimapilis**[4], **Annavi M. Villanueva**[1,4], **Jaira L. Mauhay**[1], **Motoi Suzuki**[1,2,5], **Michio Yasunami**[1], **Nobuo Koizumi**[6], **Emi Kitashoji**[2], **Kentaro Sakashita**[1], **Ikkoh Yasuda**[1,2], **Akira Nishiyama**[2], **Chris Smith**[1,7], **Koya Ariyoshi**[1,2⊕]*, **Christopher M. Parry**[1,8⊕]*

**1** School of Tropical Medicine and Global Health, Nagasaki University, Nagasaki, Japan, **2** Department of Clinical Tropical Medicine, Institute of Tropical Medicine, Nagasaki University, Nagasaki, Japan, **3** Department of Microbiology, Oita University Faculty of Medicine, Yufu, Oita, Japan, **4** San Lazaro Hospital, Manila, the Philippines, **5** Infectious Disease Surveillance Center, National Institute of Infectious Diseases, Tokyo, Japan, **6** Department of Bacteriology I, National Institute of Infectious Diseases, Tokyo, Japan, **7** Department of Clinical Research, London School of Hygiene and Tropical Medicine, London, United Kingdom, **8** Clinical Sciences, Liverpool School of Tropical Medicine, Liverpool, United Kingdom

⊕ These authors contributed equally to this work.
* kari@nagasaki-u.ac.jp (KA); christopher.parry@lstmed.ac.uk (CMP)

## Abstract

Community-acquired bacterial bloodstream infections are caused by diverse pathogens with changing antimicrobial-resistance patterns. In low-middle income countries in Southeast Asia, where dengue fever is endemic and a leading cause of fever, limited information is available about bacterial bloodstream infections due to challenges of implementing a blood culture service. This study describes bacterial bloodstream pathogens and antimicrobial-resistance patterns in Metro Manila, the Philippines. We aimed to identify the proportion of patients with a positive blood culture, the bacteria isolated and their antimicrobial resistance patterns, and the clinical characteristics of these patients, in this dengue endemic area. We conducted a prospective observational study in a single hospital enrolling febrile patients clinically suspected of having a community-acquired bacterial bloodstream infection between 1st July 2015 and 30th June 2019. Each patient had a blood culture and additional diagnostic tests according to their clinical presentation. We enrolled 1315 patients and a significant positive blood culture was found in 77 (5.9%) including *Staphylococcus aureus* (n = 20), *Salmonella enterica* Typhi (n = 18), *Escherichia coli* (n = 16), *Streptococcus pneumoniae* (n = 3) and *Burkholderia pseudomallei* (n = 2). Thirty-four patients had meningococcal disease diagnosed by culture (n = 8) or blood PCR (n = 26). Additional confirmed diagnoses included leptospirosis (n = 177), dengue virus infection (n = 159) and respiratory diphtheria (n = 50). There were 79 (6.0%, 95%CI 4.8%−7.4%) patients who died within 28 days of enrollment. Patients with a positive blood culture were significantly more likely to die than patients with negative culture (15.2% vs 4.4%, P<0.01). Among *S. aureus* isolates, 11/20 (55%) were methicillin-resistant (MRSA) and ST30: USA1100 was dominant sequence type

**Data Availability Statement:** All relevant data are within the manuscript and its Supporting Information files.

**Funding:** The work was supported by the Ministry of Education, Culture, Sports, Science and Technology (MEXT), Government of Japan to CS, KA and CMP. The funders had no role in the study design, data collection and analysis, decision to publish or reparation of the manuscript.

**Competing interests:** I have read the journal's policy and the authors of this manuscript have the following competing interests: An automated hematology analyzer, Sysmex XN-1000, and the related reagents were supplied to this study by Sysmex corporation. The Rapidchip PCT kit was provided by Sekisui Medical to this study. KA has received research funds from Sysmex Corporation for unrelated research. The other authors have no conflict of interest in the conduct of this study.

(88.9%). Antimicrobial-susceptibility was well preserved in *S. enterica* Typhi. Among hospitalized patients with clinically suspected community-acquired bacterial bloodstream infection in Metro Manila, the Philippines, 5.9% had a blood culture confirmed infection of whom 15.6% died. *S. aureus*, including a significant number of MRSA (USA1100 clones), *S. enterica* Typhi, *E.coli* and *Neisseria meningitidis* were frequently identified pathogens.

## Author summary

Antimicrobial resistance (AMR) has been increasing among some bacterial pathogens in the community settings in Southeast Asia. Community-acquired bacterial bloodstream infections can cause significant morbidity and mortality. Data on the casual bacterial pathogens and the antimicrobial resistance patterns is limited in Metro Manila, the Philippines, which is the most densely populated area in the world and highly endemic for dengue. We performed a 4-year prospective study in one tertiary hospital in Metro Manila. Community-acquired bacterial bloodstream infections were confirmed in 77 (5.9%) of 1315 clinically suspected patients. *Staphylococcus. aureus*, including methicillin-resistant (MRSA), *Salmonella enterica* Typhi, *Escherichia coli*, *Neisseria meningitidis* and leptospirosis were common pathogens. *B. pseudomallei* was an additional important pathogen observed. Although antimicrobial-susceptibility was well preserved in *S. enterica* Typhi, the emergence of a highly virulent type of MRSA was observed among some patients. The USA1100 clone was the dominant sequence type of MRSA isolates. Our findings may help to inform physicians in selecting empirical antimicrobial treatment among patients with severe febrile illness.

## Introduction

Community-acquired bacterial bloodstream infections place a significant burden on healthcare services in low and middle-income countries and cause significant morbidity and mortality [1]. The timely administration of appropriate antimicrobial therapy in bacterial bloodstream infections is crucial for saving lives but the potential range of pathogens and high levels of antimicrobial resistance (AMR) make this challenging [2,3].

In Southeast Asia, information about the etiology and resistance patterns of community-acquired bacterial bloodstream infections is patchy. A systematic review determining the etiology of community acquired bloodstream infections between 1990 and 2010 in south and southeast Asia showed that *Salmonella enterica* serotype Typhi was the most common bacterial pathogen in adult and children [4]. Improvements in access to clean water and improved sanitation have led to a reduction in the incidence of typhoid fever in Southeast Asia [5–7]. Only two studies were available from the Philippines to include in this review both conducted in rural areas before 2000 [8,9]. Dengue fever is a leading cause of hospitalisations among patients with acute infectious diseases in Southeast Asia [10–13]. The clinical presentations of dengue may overlap with those with bacterial bloodstream infection [14]. Identification of bacteraemia is difficult without blood cultures, but establishing a blood culture service is expensive and difficult to implement in many dengue endemic areas.

In an era of a falling incidence of typhoid fever, updated data on causal pathogens of community-acquired bacterial bloodstream infections and AMR is essential. The objective of this study was to provide contemporary data concerning the etiology and AMR patterns in patients

admitted to hospital in Metro Manila with a suspected community acquired bacterial bloodstream infection. Furthermore, we aimed to determine the proportion of patients with a positive blood culture, and identify the clinical characteristics of those patients, through the collection of blood cultures in a systematic sample of patients with suspected community acquired bacterial bloodstream infection.

## Methods

### Ethics statement

Ethical approval was obtained from the Research and Ethical Review Board of San Lazaro Hospital, the Philippines (reference number: SLH-RERU-2015-005-E) and the Institutional Review Board of the Institute of Tropical Medicine, Nagasaki University, Japan (150226136–4). We obtained written informed consent from participants, or their guardians or caregivers for patients aged under 18 years of age, those who were illiterate, or were unconscious at presentation.

### Setting

We performed a prospective observational study, in San Lazaro Hospital (SLH), a 500-bed national government tertiary referral hospital for infectious diseases in Manila City, Metro Manila. Metro Manila has a population of 12.8 million and high population density of 21,000/km$^2$ [15]. The hospital provides for walk-in patients in living in the area around the hospital and referred patients with suspected infectious diseases. SLH has 10,000–15,000 yearly admissions. In 2018, the leading cause of the admission was dengue (n = 3,738) followed by pulmonary tuberculosis (TB) (2,301), pneumonia (2,284), and HIV/AIDS (765). Between December 2018 and March 2019, a huge measles outbreak occurred in the Philippines and a total of 4,325 individuals with measles were admitted [16].

### Clinical methods

A research nurse approached eligible patients attending the emergency room between 9am and 4pm from Monday to Friday each week excluding national holidays. The inclusion criteria were (i) patients aged >12 months, (ii) presenting with an acute onset of fever with a duration lasting ≤ 21 days, (iii) clinically suspected to have a community-acquired bacterial bloodstream infection, and (iv) the attending physician required hospital admission and requested a blood culture within 48 hours of admission. We included patients with an identifiable focus of infection, such as pneumonia, severe pharyngitis, or skin infection, and those without identifiable focus of infection, such as leptospirosis. When the attending physician and study team suspected the patient to have bacteraemia and therefore requested a blood culture this was defined as "clinically suspected to have a community-acquired bacterial bloodstream infection". The most likely diagnosis was selected from our diagnostic lists as an admission diagnosis by an attending physician during the enrolment. The admission diagnoses were made clinically in the ER before laboratory tests were available. We excluded patients whose the most likely diagnosis considered by the attending physician was dengue fever, a viral respiratory tract infection, a viral exanthem or viral gastroenteritis. We also excluded patients who had a hospital stay within the 30 days prior to this admission or were known to have underlying chronic disease/conditions such as TB, HIV, malignancy, autoimmune disease or immunocompromised status.

Dengue fever is the most important and common differential diagnosis of bacterial blood stream infection in this setting and the majority of febrile patients admitted to the hospital

have dengue fever. If the patient was clinically diagnosed with dengue fever as the admission diagnosis, blood culture was not requested. To confirm that bacteraemia was not being missed in this group, we enrolled the first two patients in the emergency room every Tuesday morning who were suspected to have dengue fever as a dengue control convenience sample group. We obtained a blood culture from these patients.

The demographic data, medical history, clinical findings and treatments given during admission were recorded on a standard case-report form. We assessed the presence of hemo-dynamic shock, Glasgow coma score (GCS) and the quick Sequential Organ Failure Assessment (qSOFA) score at the time of enrollment [17]. All treatments were provided by attending physicians and their medical teams. The study did not involve any clinical intervention. We conducted a telephone interview with the family to determine the outcome at 28 days if patients were discharged alive within 28 days after the admission. We compared the clinical characteristics and parameters between patients with a positive blood culture result and those with a negative result.

## Laboratory methods

All patients had a blood culture set, a complete blood count, chemistry (AST, ALT, blood urea nitrogen, creatinine), C-reactive protein (CRP), procalcitonin (PCT), Dengue Rapid diagnostic test (Dengue Duo, NS1, IgM and IgG, Standard Diagnostics, South Korea), Dengue reverse transcription-polymerase chain reaction (RT-PCR), and Leptospirosis Patoc antigen-IgM enzyme linked immunosorbent assay (ELISA). Additional diagnostic tests, such as chest X-ray, nasal/throat swab for *Corynebacterium diphtheriae* culture and PCR, or *Neisseria meningitidis* blood PCR, were performed for each patient according to clinical presentation (S1 File). We collected convalescent blood samples between 7 and 10 days after enrolment or on the day of hospital discharge if sooner.

Sample processing, isolate identification, and antimicrobial susceptibility testing (AST) were conducted at the SLH-Nagasaki collaborative laboratory in SLH. Blood was inoculated into two aerobic blood culture bottles; anaerobic culture was not performed. The BacT/ALERT automated system (Organon-Teknika Corp., Durham, N.C.), with BacT/ALERT FA Plus adult bottles for patients aged ≥7 years and BacT/ALERT PF Plus pediatric bottles if aged <7 years, until November 2017. Because of an interrupted supply of BacT/ALERT bottles, from December 2017, we used the BACTEC 9050 system (Becton Dickison, Franklin Lakes, NJ) BACTEC aerobic bottles were used for patients aged ≥7 years and BACTEC Peds Plus if aged <7 years. Blood culture bottles were inoculated with the appropriate blood volume according to the age (S1 File). The volume of blood added to the bottles was assessed in a sub-set of participants by comparing the weight of the blood culture bottle before and after blood inoculation. We categorized the blood volume as adequate, less adequate and underfilled (S1 File). All culture bottles were incubated for 5 days. Bottles flagging positive were sub-cultured onto Columbia sheep blood, chocolate, and MacConkey's agars.

Bacterial isolates were identified using a MALDI Biotyper (Bruker Daltonics, Bremen, Germany) with additional standard microbiological techniques and VITEK2 compact (bioMérieux, France) where necessary. Optochin susceptibility testing was used to distinguish *Streptococcus pneumoniae* from other alpha hemolytic streptococci. Organisms considered contaminants after review of the microbiological and clinical data were excluded from the analysis. Antimicrobial susceptibility testing was performed by the modified Kirby-Bauer disc diffusion method using Sensi-Disc (BD: Becton, Dickinson and Company, USA) with additional VITEK2 compact according to the Clinical and Laboratory Standards Institute guidelines (CLSI 2015).

When the MALDI-TOF identification was *Burkholderia thailandensis* or *pseudomallei*, *Salmonella* Paratyphi A, or *Salmonella* Typhi, DNA was extracted using QIAamp DNA Blood Mini Kit following the manufacturer's instructions and PCR identification was applied using primers described elsewhere [10,11,18]. We also performed Salmonella somatic and flagellar serotyping antisera tests to confirm *Salmonella* Paratyphi A and *Salmonella* Typhi according to the manufacturer's instructions; (Denka Sieken, Japan).

A final diagnosis was made based on laboratory findings, X-ray or clinical features. If two or more laboratory tests were positive for different pathogens, a final diagnosis was made following the S1 File.

## Data analysis

We assumed that the proportion of significant positive blood cultures would be about 8% [4]. We planned to recruit 1250 patients which would result in 100 patients with a significant bacteraemia. We considered this would be a reasonable sample to understand the range of pathogens causing bacteraemia and to compare the clinical features of those with a positive blood culture with those that were negative. Clinical and laboratory data were managed in Microsoft Access 2013 (Microsoft Corporation, Redmond, WA) and statistical analyses were performed using STATA 15.0 version (Stata Corp, Texas, USA). To identify statistical difference between patients with a positive blood culture and those with a negative result, categorical variables were compared with chi-squared test or Fisher's exact test, and numerical variables by the Student's t test. To investigate the effect of various variables on the blood culture positivity, an unadjusted logistic regression model was used to show odds ratio. Values of $P < 0.05$ were considered significant. We defined November-June as the dry season and July-October as the rainy season. We report in accordance with the STROBE (Strengthening the Reporting of Observational Studies in Epidemiology) statement (S2 File).

## Results

The study period was between 1st July 2015 and 30th June 2019 inclusive. The total number of patient consultations in the ER during the study period from Monday to Friday between 9am and 4pm was 18,419 and 9,209 were admitted to the hospital. Most admitted patients were diagnosed with dengue fever, viral exanthem or viral gastroenteritis. We enrolled 1,347 patients but excluded 32 patients from the analysis because the blood culture had been taken > 48 hours after the time of admission (n = 20), the duration of the symptoms was > 21days (n = 9), blood culture was not obtainable (n = 1) and inability to obtain detailed information after enrollment (n = 2) (Fig 1). 1,315 patients were included in this analysis. We additionally enrolled 257 patients with clinically suspected dengue fever as a control group (S1 File).

The annual number of enrolled patients varied between 261 and 427. 40.8% of the patients were admitted during the rainy season (Table 1). There were 173 (13.2%) children aged between > 12 months and < 5 years, 381 (29.0%) children aged ≥ 5 years and ≤ 17 years and, 761 (57.9%) adult patients aged ≥18 years. A total of 488 (37.1%) resided in Manila City and 455 (34.6%) were referred from other health care facilities. In 311 (23.1%) patients the fever had lasted for more than 7 days and 462 (35%) patients had taken oral antibiotics prior to the consultation. There were 277 (21.1%) patients with an underlying disease, mainly asthma and diabetes, and 10 patients were pregnant. There were 69 (5.3%), patients who developed shock status during admission, 68 (5.2%) with GCS <15 and 184 (14%) with qSOFA score ≥2. The most frequent admission diagnosis was pneumonia, followed by leptospirosis, typhoid fever and severe skin infection (cellulitis or soft tissue abscess). Of 1,315 enrolled patients, 79 (6.0%, 95% confidence interval [CI],4.8%−7.4%) patients died within 28 days of enrollment.

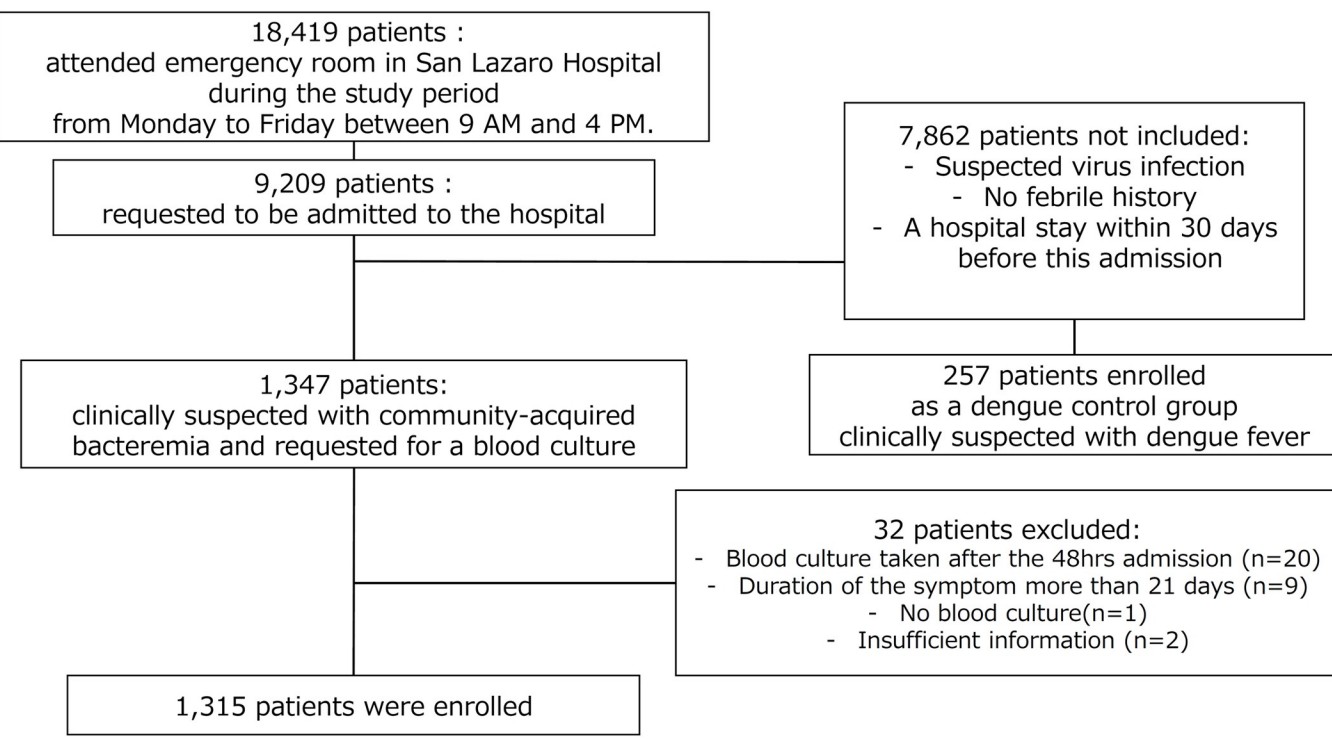

**Fig 1. Study profile and the flow of enrollment.**

A significant organism was isolated from blood culture in 77 (5.9%) of 1,315 patients and an organism considered a contaminant in 45 (3.4%). In the control group of 275 patients with clinically suspected dengue fever, the blood culture only yielded 3 (1.2%) organisms, all considered contaminants. Among the 77 patients with a significant positive blood culture, eight children were aged < 5 years and 14 were aged ≥ 5 years and ≤ 17 years. There was no significant difference in the proportion of positive blood cultures in patients with prior antibiotic use compared with those without (4.98% vs 6.33%, P = 0.39). In a sub-group of 975 patients, there was no significant difference in the proportion of positive blood cultures according to blood culture volume (S1 File). A blood culture was significantly more likely to be positive among patients presenting with coma and a low GCS, joint pain, an infected wound, vomiting, hepatomegaly, edema, joint swelling and neck stiffness (S1 File). A blood culture was also significantly more likely to be positive among patients with a high white cell count, elevated neutrophil count, high CRP, or high PCT. A positive blood culture was associated with an admission diagnosis of typhoid fever, a severe skin infection, meningococcal disease, biliary tract infection or intra-abdominal infection, and endocarditis compared with those diagnosed with pneumonia. Patients with a positive blood culture were significantly more likely to die within 28 days of admission than patients with negative blood culture (12/79 (15.2) % vs 65/1236 (4.4%), P<0.01) (Table 1).

The most common pathogen among the positive blood cultures was *Staphylococcus. aureus* (20) followed by *S. enterica* Typhi (18), *Escherichia coli* (16) and *Neisseria meningitidis* (8) (Table 2). Two patients grew *B. pseudomallei* and both died. The case fatality among patients with *N. meningitidis* was 3/8 (37.5%) and *S. aureus* 3/20 (15%).

Eleven (55%) of the 20 isolates of *S. aureus* were methicillin-resistant (MRSA) (Table 3). The PVL gene was detected in 10/11 (90.9%) of MRSA isolates and 5/9 (55.6%) of MSSA.

**Table 1. Demographic and clinical characteristics, laboratory findings and admission diagnosis among the enrolled patients at the time of admission.** Associations between the characteristics and blood culture positive results.

| | | Enrolled patients N (%) (N = 1315) | Patients N with positive blood culture (N = 77) | Blood culture positivity (%) | P value | Odds ratio (95% confidence interval) |
|---|---|---|---|---|---|---|
| Year and month of the admission | | | | | | |
| 2015July~2016Jun | | 261 (19.9) | 22 | 8.4 | 0.04 | |
| 2016July~2017Jun | | 265 (20.2) | 20 | 7.6 | | |
| 2017July~2018Jun | | 427 (32.5) | 16 | 3.8 | | |
| 2018July~2019Jun | | 362 (27.5) | 19 | 5.3 | | |
| Dry season (Nov ~ Jun) | | 778 (59.2) | 51 | 6.6 | | Ref |
| Rainy season (July ~ Oct) | | 537 (40.8) | 26 | 4.8 | 0.23 | 0.72 (0.45~1.18) |
| Age group | < 5 years | 173 (13.2) | 8 | 4.6 | | Ref |
| | 5–17 years | 381 (29) | 14 | 3.7 | 0.04 | 0.79 (0.32~1.92) |
| | 18 years and above | 761 (57.9) | 55 | 7.2 | | 1.61 (0.75~3.44) |
| Sex | Female | 500 (38) | 30 | | | Ref |
| | Male | 815 (62) | 47 | 5.8 | 0.90 | 0.96 (0.60~1.54) |
| Place of residence[a] | Manila City | 488 (37.1) | 24 | 4.9 | | Ref |
| | Outside Manila City | 826 (63.9) | 53 | 6.4 | 0.33 | 1.32 (0.81~2.17) |
| Duration of fever[a] | < = 7 days | 1003 (76.9) | 52 | 5.2 | | Ref |
| | >7 days | 311 (23.1) | 25 | 8.0 | 0.07 | 1.60 (0.97~2.62) |
| Underlying chronic condition[a] | No | 1037 (78.9) | 56 | 5.4 | | Ref |
| | Yes | 277 (21.1) | 21 | 7.6 | 0.19 | 1.44 (0.85~2.41) |
| Antibiotic use before the admission | No | 853 (64.9) | 54 | 6.3 | | Ref |
| | Yes | 462 (35.1) | 23 | 5.0 | 0.39 | 0.77 (0.47~1.28) |
| Referral from other facilities | No | 860 (65.4) | 50 | 5.8 | | Ref |
| | Yes | 455 (34.6) | 27 | 5.9 | 1.00 | 1.02 (0.63~1.65) |
| BT 37˚C> on admission | No | 477 (36.3) | 24 | 5.0 | | Ref |
| | Yes | 838 (63.7) | 53 | 6.3 | 0.39 | 1.27 (0.70~2.09) |
| Shock (mean blood pressure < 70) | No | 1246 (94.8) | 74 | 5.9 | | Ref |
| | Yes | 69 (5.3) | 3 | 4.4 | 0.79 | 0.72 (0.22~2.34) |
| Glasgow Coma Scale | = 15 | 1247 (94.8) | 66 | 5.3 | | Ref |
| | < 15 | 68 (5.2) | 11 | 16.2 | <0.01 | 3.45 (1.73~6.89) |
| qSOFA | <2 | 1131 (86) | 61 | 5.4 | | Ref |
| | > = 2 | 184 (14) | 16 | 8.7 | 0.09 | 1.67 (0.94~2.93) |
| WBC× $10^9$/L[b] | <15.0 | 1004 (77.5) | 40 | 4.0 | | Ref |
| | > = 15.0 | 292 (22.5) | 35 | 12.0 | <0.01 | 3.28 (2.04~5.27) |
| Neutrophils % [c] | <80 | 840 (64.8) | 26 | 3.1 | | Ref |
| | > = 80 | 457 (35.2) | 49 | 10.7 | <0.01 | 3.78 (2.31~6.15) |
| CRP (10> mg/dL)[d] | <10 | 866 (66.3) | 24 | 2.8 | | Ref |
| | > = 10 | 441 (66.7) | 53 | 12.0 | <0.01 | 4.79 (2.91~7.87) |
| PCT (0.75> ng/ml)[e] | <0.75 | 623 (49.1) | 11 | 1.8 | | Ref |
| | > = 0.75 | 645 (50.9) | 63 | 9.8 | <0.01 | 6.03 (3.15~11.56) |
| Outcome at 28 days | Survived | 1236 (94.0) | 65 | 4.4 | | Ref |
| | Died | 79 (6.0) | 12 | 15.2 | <0.01 | 3.23 (1.66~6.26) |
| Admission diagnosis | | | | | | |

*(Continued)*

**Table 1.** (Continued)

| | Enrolled patients N (%) (N = 1315) | Patients N with positive blood culture (N = 77) | Blood culture positivity (%) | P value | Odds ratio (95% confidence interval) |
|---|---|---|---|---|---|
| Pneumonia | 378 (28.8) | 11 | 2.9 | <0.01 | |
| Leptospirosis | 242 (18.4) | 5 | 2.1 | | |
| Typhoid fever | 118 (9.0) | 20 | 17.0 | | |
| Severe skin infection | 116 (8.8) | 13 | 11.2 | | |
| UTI | 116 (8.8) | 10 | 8.6 | | |
| Acute gastroenteritis | 94 (7.2) | 1 | 1.1 | | |
| Diphtheria | 68 (5.2) | 0 | 0 | | |
| Meningococcus | 59 (4.5) | 10 | 17.0 | | |
| CNS infection | 54 (4.1) | 2 | 3.7 | | |
| Biliary tract infection/intra abdominal infection | 22 (1.7) | 3 | 13.6 | | |
| Endocarditis | 3 (0.2) | 2 | 66.8 | | |
| Others | 45 (3.4) | 0 | 0 | | |

CRP, C-reactive protein; PCT, procalcitonin; qSOFA, quick Sequential Organ Failure Assessment;

[a] 1missing data,

[b] 19 missing value

[c] 18 missing value

[d] 8 missing value

[e] 47 missing value

MLST was determined for 16 isolates (9 MRSA, 7 MSSA). ST 30 was dominant ST among the MRSA 8/9 (88.9%) and one isolate was ST 508. MRSA isolates carried a specific type of SCC*mec* identified as a subtype of SCC*mec* IV. Among the 20 cases with *S. aureus* bacteremia, 12 cases had severe skin infections. The isolates of *S. enterica* Typhi were susceptible to all the tested antimicrobials. *E. coli* susceptibility to common antimicrobials was also well preserved. Only one isolate of *E. coli* had an extended spectrum Beta-Lactamase (ESBL) phenotype and no MDR or XDR strains were found.

Thirty-four patients were diagnosed with meningococcal infection confirmed by blood culture in 8 (23.5%) and by blood meningococcal PCR 26 (76.5%). PCR serotyping in 24 PCR or blood culture positive samples showed serogroup B in 18 (75%), serogroup Y in 3 (12.5%) and

**Table 2. Bloodstream isolates among enrolled patients with the case fatality and the age group.**

| | Total | < 5 years | 5 – 17 years | ≥18 years | Died | Case fatality (%) |
|---|---|---|---|---|---|---|
| *Staphylococcus aureus* | 20 | 3 | 3 | 14 | 3 | 15 |
| *Salmonella enterica* Typhi | 18 | 1 | 7 | 10 | 0 | 0 |
| *Escherichia coli* | 16 | 0 | 1 | 15 | 1 | 6.25 |
| *Neisseria meningitidis* | 8 | 3 | 2 | 3 | 3 | 37.5 |
| *Streptococcus pneumoniae* | 3 | 0 | 0 | 3 | 0 | 0 |
| *Burkholderia pseudomallei* | 2 | 0 | 0 | 2 | 2 | 100 |
| *Klebsiella pneumoniae* | 2 | 0 | 0 | 2 | 1 | 50 |
| *Proteus mirabilis* | 2 | 0 | 0 | 2 | 0 | 0 |
| *Pseudomonas aeruginosa* | 2 | 0 | 0 | 2 | 2 | 100 |
| *Non-typhoidal salmonellae* | 2 | 0 | 0 | 2 | 0 | 0 |
| *Enterobacter cloacae* | 1 | 0 | 1 | 0 | 0 | 0 |
| *Streptococcus pyogenes* | 1 | 1 | 0 | 0 | 0 | 0 |

**Table 3. Antimicrobial susceptibility pattern of isolates from enrolled patients.**

| | Number of susceptible isolates / Number of tested isolates | | | |
|---|---|---|---|---|
| | *S. aureus* | *S. enterica* Typhi | *E. coli* | *N. meningitidis* |
| Total N. of isolates | **20 (MRSA n = 11)** | **18** | **16 (ESBL n = 1)** | **8** |
| PenicillinG | 0/19 | | | |
| Ampicillin | | 18/18 | 3/16 | |
| Amoxicillin_clavulanate | | | 7/16 | |
| Piperacillin | | | 5/16 | |
| Piperacillin_Tazobactam | | | 16/16 | |
| Aztreonam | | | 11/12 | |
| Cefoxitin | 9/20 | | 7/7 | |
| Ceftazidime | | | 15/16 | |
| Ceftriaxone | | 18/18 | 15/16 | 8/8 |
| Cefepime | | | 15/16 | |
| Imipenem | | | 16/16 | |
| Meropenem | | | 16/16 | 7/7 |
| Ciprofloxacin | 19/20 | 18/18[a] | 14/16 | 8/8 |
| Levofloxacin | 19/19 | | 14/16 | |
| Nalidixic_Acid | | 18/18 | | |
| Azithromycin | | 18/18[b] | | |
| Clindamycin | 19/20 | | | |
| Erythromycin | 19/20 | | | |
| Gentamicin | 19/20 | | 14/16 | |
| Amikacin | | | 16/16 | |
| Linezolid | 18/18 | | | |
| Rifampin | 20/20 | | | 8/8 |
| Tetracycline | 19/20 | | 4/11 | |
| Trimethoprim-sulfamethoxazole | 19/20 | 18/18 | 6/15 | 0/8 |
| Chloramphenicol | 15/16 | 18/18 | 14/16 | 8/8 |
| Vancomycin | 20/20 | | | |

[a]E-test MIC 0.003~0.016

[b]E-test MIC:0.03~6

undetermined in three. *N. meningitidis* susceptibility to common antimicrobials was also well preserved.

A causal pathogen was identified by laboratory tests in 454 (34.5%) of the 1,315 patients (Table 4). Diagnostic tests were positive for dengue virus in 159 (12.1%) and leptospirosis in 177 (13.5%). Five patients with primary diagnosis of leptospirosis were also blood culture positive (*E. coli* (3), *P. aeruginosa* (1), Non-typhoidal *Salmonellae* (1)). Fifty patients were confirmed to have respiratory diphtheria based on culture or PCR laboratory tests, but none were blood culture positive.

## Discussion

In this prospective study of Filipino patients admitted to hospital with a suspected community acquired bacterial bloodstream infection, a blood culture was positive in 5.9% of patients. Common pathogens were *S. aureus*, *S. enterica* Typhi, *E. coli*, and *N. meningitidis*. The case fatality rate at 28 days in all patients was 6.0% (95% CI, 4.8%−7.4%) but was significantly higher at 15.2% in patients with a positive blood culture. Previous prospective studies of blood

**Table 4. Final diagnosis based on Laboratory confirmation, X-ray and clinical diagnosis of severe skin infection.**

| Diagnosis | Suspected bacterial infection N (%) N = 1315 | Suspected Dengue infection controls N (%) N = 257 |
|---|---|---|
| Bacteremia | 77 (5.9) | 0 (0) |
| Proven Dengue {Dengue NS1 (+) or RT-PCR (+)} | 79 (6.0) | 98 (38.1) |
| Probable Dengue possible {NS1(-) and RT-PCR(-)} and {RDT IgM(+) or ELISA IgM(+)} | 80 (6.1) | 48 (18.7) |
| Proven Leptospirosis {PCR(+) or Culture(+) or IgM seroconversion(+)} | 97 (7.4) | 1 (0.4) |
| Probable Leptospirosis Lepto {PCR(-) or Culture (-)} and IgM (+) | 45 (3.4) | 0 (0) |
| Xray confirmed Pneumonia (Blood culture negative)[a] | 86 (6.5) | 0 (0) |
| Diphtheria | 50 (3.8) | 0 (0) |
| Meningococcus (Blood culture negative)[b] | 26 (2.0) | 0 (0) |
| Severe Skin infection (Blood culture negative)[c] | 59 (4.5) | 0 (0) |
| No diagnosis confirmed | 716 (54.4) | 110 (42.8) |

[a]Blood culture positive pneumonia (N = 8)

[b]Blood culture positive meningococcus (N = 8)

[c] Blood culture positive skin infection (N = 15). These diagnoses are based on all the available results

stream infection in participants admitted to hospital with fever or sepsis in Southeast Asia are summarized in Table 5 [10–12,18–20]. The proportion of positive blood cultures in our study was less than the 8% expected but consistent with the 2.2% and 10.5% range seen in these studies. Prior antibiotic use and low blood volumes were not associated with negative blood cultures. We identified several clinical signs, blood tests and the clinical diagnosis associated with positive blood cultures.

*S. aureus* was the leading cause of bloodstream infection, with a high rate of MRSA and PVL positive isolates causing skin and soft tissue infections. ST30 (USA1100) was the most prevalent sequence type among MRSA isolates. Several reports identify ST30 infections in other Southeast Asian countries and in isolates originating from the Philippines [21–25]. A multinational surveillance in Asian countries between 2004 and 2006 showed the proportion of MRSA in the Philippines was lower than other countries and ST30 was found from CA-MRSA and HA-MRSA isolates [26]. In another study, in a private hospital in Manila city 41/108 (38%) of MRSA isolates were positive for PVL [27].

Typhoid was the second commonest cause of community acquired bloodstream infection but was less prominent as a cause compared with other studies in Southeast Asia (Table 5)[10–12,18–20]. *S. enterica* Typhi isolates were susceptible to all antimicrobials tested in contrast to other Asian countries [28]. Although, imported cases of the ESBL-producing *S. enterica* Typhi from Philippines have been described [29,30], national data and previous reports show preserved susceptibilities [31–33]. The antimicrobial susceptibilities of other Enterobacterales, including *E. coli* and *Klebsiella. pneumoniae*, were also preserved in our study although the number of isolates was small. National surveillance reports demonstrate high rates of resistance to extended spectrum cephalosporins in the Philippines. Ceftriaxone resistance of 460/1187 (38.8%) in *E. coli* and 737/1363 (54.1%) in *K. pneumoniae* were reported in 2020 [33].

Our study detected only three isolates of *S. pneumoniae* and none of *Haemophilus. influenzae*. Pneumococcal conjugate vaccine (PCV) and *Haemophilus influenzae* type b (Hib) vaccine were introduced in 2014 and 2010 respectively. Although vaccine coverage is not high, at around 35% and 67% between 2014 and 2019, respectively [34], the vaccine may still influence

**Table 5. Common etiologies detected by blood culture and other laboratory methods in previous studies in Southeast Asian countries.**

| Study site | Year the study conducted | Target population | Enrollment criteria | N of BC positive / Total N (BC Positivity %) | Isolates detected by BC; N (%) [Antimicrobial Resistant (%)] | Other identified etiology N (%) |
|---|---|---|---|---|---|---|
| Myanmar [18] | Oct 2015 – Oct 2016 | Adolescent (12>years) and adults | Fever ≥ 38˚C | 90[a] / 947 (9.5) | *S.* Typhi; 33 (36.3) [Nalidixic acid (100), Ciprofloxacin (100), Azithromycin (0), Cefriaxon (0)], *E. coli*; 20 (22.0) [ESBL (75), MDR (85), XDR (10)], *S.* Paratyphi A; 10 (11.0), *K. pneumonia*; 7 (7.7) [ESBL (43), MDR (43)], *S. aureus*; 6 (6.6) [MRSA (0)], *B. pseudomallei*; 1 (1.1) | NR |
| Indonesia, Thailand, Vietnam [11] | Dec 2013-Dec 2015 | children (age ≥30 days) and adults | Sepsis (1578) | 131 / 1531 (8.6) | *E. coli*; 40 (30.5), -*S. aureus*; 21 (25.2), *S. pneumoniae*; 10 (7.6), *K. pneumoniae*; 9 (6.9), *Acinetobacter* spp; 9 (6.9), *B. pseudomallei*; 3 (2.3%), *S.* Typhi; 1 (0.8%) | Dengue viruses; 122 (8), *Leptospira* spp; 95 (6), Rickettsial pathogens; 96 (6), Influenza virus; 65 (4), *Plasmodium* spp; 12 (1) |
| Laos [10] | May 2008-Dec 2010 | Children and adult (5–49 years) | fever | 43 / 1938 (2.2) | *S.* Typhi; 38 (72%) [Nalidixic acid (2.6), Ceftriaxone (0)], *E. coli*; 4 (8%) [Ceftriaxone (25)], *B. pseudomallei*; 3 (6), *S. aureus*; 2 (4), *K. pneumoniae*; 2 (4) | Dengue virus;156 (8), Scrub typhus; 122 (7), Japanese encephalitis virus; 112 (6), *Leptospira* spp; 109 (6) |
| Cambodia [19] | July 2007-Dec 2010 | Adult (15–99 years) | SIRS | 445 / 4833 (9.2) [b] | *E. coli*; 132 (29.7%) [ESBL (47.7)], *B. pseudomallei*; 56 (12.6), *S. aureus*; 53 (11.9%)[c] [MRSA (21.7)], *K. pneumoniae*; 34 (7.6) [ESBL (43.8)], *S.* Typhi; 15 (3.4) [Ciprofloxacin (90.0), Azithromycin (5.0)], *S.* Paratyphi *A*; 2 (0.4) | NR |
| Cambodia [12] | Oct 2009-Oct 2010 | Children (<16 years) | Fever, admitted to hospital | 76 / 1212 (6.3) | *S.* Typhi; 22 (1.8)[Ciprofloxacin (90.5), MDR (85.7)], *S. pneumoniae*; 13 (1.1%) [Ceftriaxone (0%)], *E. coli*; 8 (0.7%) [ESBL (50)], *S. aureus*; 6 (0.5%) [MRSA (17.7)], *B. pseudomallei*; 6 (0.5), *K. pneumoniae*; 4 (0.3) [ESBL (50)] | Dengue virus; 198 (16.2), Rickettsial pathogens; 134 (10.9), Japanese encephalitis virus;71 (5.8), *Leptospira* spp; 17 (1.4), Plasmodium spp; 24 (1.9%) |
| Laos [20] | Feb 2000– 2004 | Children and adult (0–100 years) | Suspected community-acquired bacteremia | 483 / 4512 (10.7) | *S.* Typhi 246 (50.9) [Nalidixic acid (0), Ceftriaxone (0)], *S. aureus*; 92 (19.0) [MRSA (0)], *E. coli*; 60 (12.4), *K. pneumoniae*; 20 (4.1), *B. pseudomallei*; 14 (2.9), *P. aeruginosa*; 14 (2.9) | NR |
| Philippines (current study) | July 2015 – Jun 2019 | Children (>12months) and adult | Clinically suspected bacteremia | 77 / 1315 (5.9) | *S. aureus*; 20 (26.0) [MRSA (55)], *S.* Typhi; 18 (23.4) [Nalidixic acid (0), Ciprofloxacin (0), Azithromycin (0), Cefriaxone (0)], *E. coli;* 16 (20.8) [ESBL (6.2)], *N. meningitidis*: 8 (10.4), *S. pneumoniae;* 3 (3.9), *B. pseudomallei*; 2 (2.6), *K. pneumoniae*; 2 (2.6) | Dengue virus; Proven 79 (6%), Proven + Probable 156 (11.9), *Leptospira* spp; Proven 97 (7.4), Proven+Probable 142 (10.8), *N. meningitidis* (BC+PCR): 34 (2.6), Diphtheria 50 (3.8) |

NR: Not Reported, CA: Community-acquired, BC: Blood culture, SIRS: Systemic inflammatory response syndrome, MRSA: Methicillin-resistant Staphylococcus aureus, ESBL: Extended spectrum beta-lactamase, PCR: Polymerase chain reaction

First we searched PubMed using the query and terms: (((sepsis[MeSH Terms]) OR (bcteremia[MeSH Terms])) AND (southeast asia[MeSH Terms])) AND (("2000/1/1"[Date—Publication]: "2020/12/31"[Date—Publication])). Then we did manual review to find the study with the inclusion criteria: (1) mainly community-acquired infection (2) Prospective analysis (3) Blood cultures were taken from more than 500 patients

[a] 6

[b] 41 and

[c] 5 cases were healthcare-associated infections.

the low incidence of bacteremia. We found 34 laboratory confirmed meningococcal infections including 26 patients positive by PCR but negative by blood culture. Serogroup B was the most common serogroup detected in our study. There is limited data about the common serogroups among Filipinos although one study investigating carriage showed that serogroup B was the most common [35,36]. Meningococcal serogroup B vaccine is not available in the Philippines.

Respiratory diphtheria can present as a critical illness among children in the Philippines and physicians order blood culture to rule out other severe bacterial infections [37]. All blood cultures were negative in this study. We identified 142 (10.8%) cases of probable and proven leptospirosis. During the rainy season in Manila many people are exposed to environmental contaminated water [38]. There were two cases of melioidosis caused by *B. pseudomallei*, one residing in Manila city and the other in a city beside Metro Manila. Melioidosis is probably under reported in the Philippines because of lack of awareness and limited diagnostic laboratory capacity [39].

We identified several clinical signs and blood results that were associated with a positive blood culture. Of note, the sample size (n = 1315) had an adequate power (>80%) if there was more than a 12% difference in the presence of a parameter between the blood culture positive and a negative groups. Our analysis is limited because we did not correct for multiple comparisons and did not perform multivariable analysis to adjust ORs. Further analysis is necessary to identify features that indicate in which patients it would be useful to take a blood culture.

Our inclusion criteria were designed to capture patients with community acquired bacterial bloodstream infections. If the criteria had been based on fever alone, the proportion of dengue patients would be very high. Blood cultures were all negative among patients enrolled as a control group with primary diagnosis of dengue confirming that excluding this group did not lead to missed patients. It is noteworthy that dengue was still a significant cause of fever among patients these patients with suspected bacterial bloodstream infections even after we excluded clinically diagnosed dengue patients. Our study only enrolled patients during the daytime and weekday at a single tertiary infectious diseases referral hospital in Metro Manila. The hospital is located in an area with a high population density and a lower economic status than the average in the Philippines and therefore may not be fully representative subjects in the Philippines [40]. We used IgM Leptospirosis ELISA as a confirmatory test for proven or probable leptospirosis, although the test can cause false positive results [41]. We did not systematically test for HIV infection so our study may have included patients with undiagnosed HIV infection. We did not perform laboratory tests for rickettsiosis. The burden of rickettsiosis is largely unknown and may be underestimated [42].

## Conclusion

Systematic investigation of patients admitted to hospital in Metro Manila, the Philippines with suspected community-acquired bacterial bloodstream infection detected a low number of positive blood cultures. CA-MRSA (ST30; USA1100), *S. enterica* Typhi *E. coli* and *N. menigitidis* were found to be the leading bloodstream infections. *B. pseudomallei* was an additional important pathogen observed. Performing blood cultures is expensive and technically demanding. In a resource-limited setting, criteria for the selection of patient groups that most benefit from blood culture would be useful.

## Supporting information

**S1 File. Supporting material.** This supporting material includes additional study protocol, laboratory method, characteristics of control dengue patients, other supporting data. (PDF)

**S2 File. STROBE Statement and the checklist of the study.**
(PDF)

**S1 Data. Data file of the study.**
(XLSX)

## Acknowledgments

We appreciate that an automated hematology analyzer, Sysmex XN-1000, and the related reagents were supplied to this study by Sysmex Corporation. The Rapidchip PCT kit was provided by Sekisui Medical to this study.

## Author Contributions

**Conceptualization:** Nobuo Saito, Rontgene M. Solante, Motoi Suzuki, Michio Yasunami, Chris Smith, Koya Ariyoshi, Christopher M. Parry.

**Data curation:** Nobuo Saito.

**Formal analysis:** Nobuo Saito, Koya Ariyoshi, Christopher M. Parry.

**Funding acquisition:** Chris Smith, Koya Ariyoshi, Christopher M. Parry.

**Investigation:** Nobuo Saito, Rontgene M. Solante, Ferdinand D. Guzman, Elizabeth O. Telan, Dorcas V. Umipig, Carina H. Frayco, Jezreel C. Lazaro, Maricel R. Ribo, Alexis Q. Dimapilis, Virginia O. Dimapilis, Annavi M. Villanueva, Jaira L. Mauhay.

**Methodology:** Nobuo Saito, Motoi Suzuki, Michio Yasunami, Nobuo Koizumi, Emi Kitashoji, Kentaro Sakashita, Ikkoh Yasuda, Akira Nishiyama, Chris Smith, Koya Ariyoshi, Christopher M. Parry.

**Project administration:** Nobuo Saito, Rontgene M. Solante, Ferdinand D. Guzman, Elizabeth O. Telan, Dorcas V. Umipig, Joy P. Calayo, Carina H. Frayco, Jezreel C. Lazaro, Maricel R. Ribo, Alexis Q. Dimapilis, Virginia O. Dimapilis, Annavi M. Villanueva, Jaira L. Mauhay.

**Resources:** Rontgene M. Solante, Ferdinand D. Guzman, Elizabeth O. Telan, Dorcas V. Umipig, Carina H. Frayco, Jezreel C. Lazaro, Maricel R. Ribo, Alexis Q. Dimapilis, Virginia O. Dimapilis.

**Software:** Nobuo Saito.

**Supervision:** Nobuo Saito, Motoi Suzuki, Michio Yasunami, Nobuo Koizumi, Emi Kitashoji, Kentaro Sakashita, Ikkoh Yasuda, Chris Smith, Koya Ariyoshi, Christopher M. Parry.

**Validation:** Nobuo Saito, Nobuo Koizumi, Chris Smith, Koya Ariyoshi, Christopher M. Parry.

**Visualization:** Nobuo Saito.

**Writing – original draft:** Nobuo Saito, Chris Smith, Koya Ariyoshi, Christopher M. Parry.

**Writing – review & editing:** Nobuo Saito, Rontgene M. Solante, Ferdinand D. Guzman, Elizabeth O. Telan, Dorcas V. Umipig, Carina H. Frayco, Jezreel C. Lazaro, Maricel R. Ribo, Alexis Q. Dimapilis, Virginia O. Dimapilis, Annavi M. Villanueva, Jaira L. Mauhay, Motoi Suzuki, Michio Yasunami, Nobuo Koizumi, Emi Kitashoji, Kentaro Sakashita, Ikkoh Yasuda, Akira Nishiyama, Chris Smith, Koya Ariyoshi, Christopher M. Parry.

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
