## [Decision Letter · Decision Letter 0]

20 Jan 2022

Dear Dr Saito,

Thank you very much for submitting your manuscript "A prospective observational study of community-acquired bacterial bloodstream infections in Metro Manila, the Philippines" for consideration at PLOS Neglected Tropical Diseases. Your paper was reviewed by members of our editorial team and by one peer reviewer. While we typically will not proceed with a single peer reviewer, after many months we have not been able to get any additional reviewers and do not want to hold up this manuscript any longer. In light of the reviews (below this email), we would like to invite the resubmission of a significantly-revised version that takes into account the reviewers' comments. 

We cannot make any decision about publication until we have seen the revised manuscript and your response to the reviewers' comments. Your revised manuscript is also likely to be sent to reviewers for further evaluation.

Sincerely,

Andrew S. Azman

Deputy Editor

Reviewer's Responses to Questions

**Key Review Criteria Required for Acceptance?**

**Methods**

-Are the objectives of the study clearly articulated with a clear testable hypothesis stated?

-Is the study design appropriate to address the stated objectives?

-Is the population clearly described and appropriate for the hypothesis being tested?

-Is the sample size sufficient to ensure adequate power to address the hypothesis being tested?

-Were correct statistical analysis used to support conclusions?

-Are there concerns about ethical or regulatory requirements being met?

Reviewer #1: -Are the objectives of the study clearly articulated with a clear testable hypothesis stated?

While this was mainly a descriptive study of patients with bloodstream infection, and the objectives are explained, there are no hypotheses clearly stated. Yet, statistical tests with p values etc are presented in the results, so hypothesis testing is implied. I think these should be more clearly articulated or it should be clarified that this was a purely descriptive study and no hypotheses are being examined. However, there are clearly some hypotheses being tested here - eg. whether BC positivity changes over time, by seasons, by clinical variables etc 

-Is the study design appropriate to address the stated objectives?

Yes, the design is broadly appropriate. I think the choice of control group may need some further explanation. Why only dengue positive cases were chosen, for instance? Why this sample size for the control population, or was it just a convenience sample? It seems the point of the control group was to demonstrate that positive blood cultures are not being missed in patients with other febrile syndromes 

(editor comment) Ln 116: What does “clinically suspected to have a bacterial bloodstream infection” mean exactly?

-Is the population clearly described and appropriate for the hypothesis being tested?

Yes

-Is the sample size sufficient to ensure adequate power to address the hypothesis being tested?

No formal sample size calculations are presented - and this ties back to the lack of clear hypotheses. While the sample size is probably reasonable for this kind of descriptive study, questions of power adequacy are hard to answer as it is not clear what precisely is being tested. 

-Were correct statistical analysis used to support conclusions?

A few potential issues to note:

a) I am not sure that ORs to compare the year of admission (using 2015-6 as a baseline reference) and the rate of BC positivity is the best way to do this. I would have thought that a test for temporal trends might be more appropriate (although I am not a statistician so seeking advice on this would be recommended). I wonder whether it might be more informative to present these graphically in some way, which may also show seasonal variation as well as year-year trends? 

b) there are mulitple ORs calculated for a number of clinical and demographic variables - was there any adjustment for multiplicity of testing? Is there not a risk that some associations will thrown up by chance alone? 

c) It was a little unclear with the "initial" clinical diagnosis if several of these were made on the basis of the blood culture results (e.g. meningococcus, endocarditis), in which case comparing the likelihood of BC positivity seems confounded and not very informative (and again comparing ORs to pneumonia as a baseline condition seems like an unusual way to present things)?

-Are there concerns about ethical or regulatory requirements being met?

None

**Results**

-Does the analysis presented match the analysis plan?

-Are the results clearly and completely presented?

-Are the figures (Tables, Images) of sufficient quality for clarity?

Reviewer #1: -Does the analysis presented match the analysis plan?

The presentation of multiple ORs in Table 1 is not really explained or justified in the statistical methods (see above)

-Are the results clearly and completely presented?

Otherwise OK - see above about presentation and analysis of the seasonal / time trends 

They state that no MDR/XDR E coli strains were found, but at least one was meropenem resistant in Table 3 - are the authors sure there was no carbapenem-resistant E coli, which would normally be at least MDR if not XDR? Some further clarity here would be useful. 

-Are the figures (Tables, Images) of sufficient quality for clarity?

Yes

**Conclusions**

-Are the conclusions supported by the data presented?

-Are the limitations of analysis clearly described?

-Do the authors discuss how these data can be helpful to advance our understanding of the topic under study?

-Is public health relevance addressed?

Reviewer #1: -Are the conclusions supported by the data presented?

Overall yes

-Are the limitations of analysis clearly described?

I think it would be worth stating that Leptospira IgM positivity alone (or even seroconversion with confirmatory testing like MAT) can have false positives. 

-Do the authors discuss how these data can be helpful to advance our understanding of the topic under study?

Yes - present useful data overall 

-Is public health relevance addressed?

Yes and placed into context of other similar studies in the region

**Editorial and Data Presentation Modifications?**

Reviewer #1: Few minor typos etc:

1, Table 5: check spelling of species as some errors e.g. "Acinetoaceter", "B. pseudomalle" "S pneumonia"

2. Author summary "Melioidosis" is a disease, not a pathogen - suggest "Burkholderia pseudomallei" is referring to the organism

**Summary and General Comments**

Reviewer #1: Overall I think the study presents some useful data, and the authors should be commended on the amount of work it must have taken to collect the information. My main concerns relate to a clearer articulation of the hypothesis and presentation of the analysis, but publication would be worthwhile if these can be addressed.

PLOS authors have the option to publish the peer review history of their article (what does this mean?). If published, this will include your full peer review and any attached files.

Reviewer #1: No
---

## [Decision Letter · Decision Letter 1]

12 Apr 2022

Dear Dr Saito,

We are pleased to inform you that your manuscript 'A prospective observational study of community-acquired bacterial bloodstream infections in Metro Manila, the Philippines' has been provisionally accepted for publication in PLOS Neglected Tropical Diseases.

Best regards,

Andrew Azman

Deputy Editor

Reviewer's Responses to Questions

**Key Review Criteria Required for Acceptance?**

**Methods**

-Are the objectives of the study clearly articulated with a clear testable hypothesis stated?

-Is the study design appropriate to address the stated objectives?

-Is the population clearly described and appropriate for the hypothesis being tested?

-Is the sample size sufficient to ensure adequate power to address the hypothesis being tested?

-Were correct statistical analysis used to support conclusions?

-Are there concerns about ethical or regulatory requirements being met?

Reviewer #1: yes

**Results**

-Does the analysis presented match the analysis plan?

-Are the results clearly and completely presented?

-Are the figures (Tables, Images) of sufficient quality for clarity?

Reviewer #1: yes

**Conclusions**

-Are the conclusions supported by the data presented?

-Are the limitations of analysis clearly described?

-Do the authors discuss how these data can be helpful to advance our understanding of the topic under study?

-Is public health relevance addressed?

Reviewer #1: yes

**Editorial and Data Presentation Modifications?**

Reviewer #1: No additional comments

**Summary and General Comments**

Reviewer #1: Thank you for the responses to the earlier comments. I am satisfied with the revised manuscript and would recommend acceptance at this stage

PLOS authors have the option to publish the peer review history of their article (what does this mean?). If published, this will include your full peer review and any attached files.

Reviewer #1: No

---

## [Editor Report · Acceptance letter]

20 May 2022

Dear Dr Saito,

We are delighted to inform you that your manuscript, "A prospective observational study of community-acquired bacterial bloodstream infections in Metro Manila, the Philippines," has been formally accepted for publication in PLOS Neglected Tropical Diseases.

Best regards,

Shaden Kamhawi

co-Editor-in-Chief

Paul Brindley

co-Editor-in-Chief
